# Reliability and Validity of a Wearable Sensing System and Online Gait Analysis Report in Persons after Stroke

**DOI:** 10.3390/s23020624

**Published:** 2023-01-05

**Authors:** Anne Schwarz, Adib Al-Haj Husain, Lorenzo Einaudi, Eva Thürlimann, Julia Läderach, Chris Awai Easthope, Jeremia P. O. Held, Andreas R. Luft

**Affiliations:** 1Vascular Neurology and Neurorehabilitation, Department of Neurology, University of Zurich, 8091 Zurich, Switzerland; 2Cereneo Foundation, Center for Interdisciplinary Research (CEFIR), 6354 Vitznau, Switzerland; 3Rehabilitation Center Triemli Zurich, Valens Clinics, 8063 Zurich, Switzerland; 4Cereneo, Center for Neurology and Rehabilitation, 6354 Vitznau, Switzerland

**Keywords:** stroke, gait rehabilitation, gait analysis, inertial measurement unit

## Abstract

The restoration of gait and mobility after stroke is an important and challenging therapy goal due to the complexity of the potentially impaired functions. As a result, precise and clinically feasible assessment methods are required for personalized gait rehabilitation after stroke. The aim of this study is to investigate the reliability and validity of a sensor-based gait analysis system in stroke survivors with different severities of gait deficits. For this purpose, 28 chronic stroke survivors (9 women, ages: 62.04 ± 11.68 years) with mild to moderate walking impairments performed a set of ambulatory assessments (3× 10MWT, 1× 6MWT per session) twice while being equipped with a sensor suit. The derived gait reports provided information about speed, step length, step width, swing and stance phases, as well as joint angles of the hip, knee, and ankle, which we analyzed for test-retest reliability and hypothesis testing. Further, test-retest reliability resulted in a mean ICC of 0.78 (range: 0.46–0.88) for walking 10 m and a mean ICC of 0.90 (range: 0.63–0.99) for walking 6 min. Additionally, all gait parameters showed moderate-to-strong correlations with clinical scales reflecting lower limb function. These results support the applicability of this sensor-based gait analysis system for individuals with stroke-related walking impairments.

## 1. Introduction

The gait or balance impairments affect approximately 50–80% of all stroke survivors [1,2], with only 30–50% of them being able to walk in the community 6 months after stroke [3]. The stroke-related gait deficits are due to hemiparesis, which leads to reduced stride frequency and step length, a longer stance on the less-affected side, prolonged swings, and decreased knee and hip flexion on the affected side [4]. In addition, other characteristics of post-stroke gait are reduced weight shift to the affected side [5], delayed postural reactions [6], shift of the center of gravity to the non-paretic side, and reduced capability to avoid obstacles [7]. The gait impairments after stroke were further described in relation to cognitive impairments; these are also present in a person after stroke without paresis, a loss of coordination, or visuospatial impairments with cognitive deficits [8]. These deficits increase the risk of falling and limit daily activities, thereby reducing the quality of life [9]. Although hemiparetic gait has its characteristic appearance, each patient has an individual impairment profile.

In addition, the assessment of individual gait performance characteristics is necessary. The common clinical assessments are limited to time-, distance-, or observer-based scales, the evaluation of balance, and patient-reported outcomes. None of these assessments provides specific information on movement quality or gait parameters [10]. Therefore, comprehensive motion analysis is necessary but time-consuming and expensive in terms of training needed and data processing, and is thus not used in clinical practice [11]. Several ongoing developments in wearable motion sensors and algorithms show potential for assessing gait in different daily life situations outside the laboratory setting. Additionally, wearable motion sensors are based on accelerometry or inertial measurement units (IMUs), which combine accelerometer, gyroscope, and magnetometer data and can be attached to different body segments. Existing applications of motion sensors span from step counts, available in most of the existing smartphones [12], to motion sensors worn on the foot [13] and lower back to capture measures such as step length and width [14]. Further, IMU-based systems can be used to reconstruct movements in more extensive setups and human models, enabling the analysis of specific segment and/or joint ranges of motion. For example, a setup of 17 IMUs enables whole-body human motion analysis [15] in various settings, such as outdoor running analysis [16] or monitoring stroke rehabilitation and real-life performance after a stroke [17,18]. Recently, an application for an automated gait-specific motion analysis by fusing a suit’s sensor data and storing it through a secure cloud-based ecosystem was reported [19]. The generated gait reports contain information about specific gait cycle features based on gait event detection of foot strike and release, as well as joint motion patterns, center of mass tracking, and leg segment accelerations. The sensor-based gait parameters included in the report were validated in a motion laboratory with camera-based tracking and force plates in a sample of 35 healthy individuals, producing overall satisfying results [19]. However, the applicability and clinimetric properties of sensor-based gait analysis systems in the stroke population are unknown. In order to be of clinical use, the system should provide reliable and valid gait analysis data in an easy-to-use and time-efficient manner [11].

Furthermore, the main objective of this study was to determine the reliability and validity of gait kinematics measured with a wearable gait analysis system during clinical ambulation tests in chronic stroke subjects. First, we aimed to investigate the usability in terms of the training needed to operate the system and the cumulative time needed to set up the system, record gait, and process the report. The second goal was to determine the test-retest reliability of the gait report parameters in different ambulation test conditions. The third aim was to investigate the construct validity by means of hypothesis testing between the gait report kinematics and clinical scales on the strength, mobility, and physical activity in a person after stroke. Finally, we tested the discriminability between the affected and the less-affected lower limbs considering clinical test conditions. The results of this study will help determine the utility and limitations of sensor-based gait analysis in stroke rehabilitation.

## 2. Materials and Methods

In order to investigate the clinimetric properties of a wearable and online-based gait analysis system in people with stroke-related gait impairments, we recruited people post-stroke, at least 6 months after the event, for one measurement time point at the University Hospital of Zurich (Department of Neurology, Division of Neurorehabilitation) in Zurich, Switzerland. The eligible participants had to be able to walk without physical assistance for at least 10 m (Functional Ambulation Categories ≥ 3). In addition, the respondents were excluded from study participation if they had other gait impairments unrelated to the stroke and/or if non-compliance was expected. The ethical clearance certificate for this study (BASEC-No: 2019-00565) was approved by the cantonal ethics committee (Ethikkommission Zürich, Zürich, Switzerland). All participants provided written consent after being informed about the study procedures.

### 2.1. Measurement System

The wearable 3D motion sensor suit consisted of 17 IMUs that were attached to predefined anatomic landmarks (head, sternum, sacrum, upper arms, forearms, hands, shoulders, upper legs, lower legs, and feet) of the subject to be analyzed. Each IMU consisted of an accelerometer, a gyroscope, and a magnetometer that measure with a latency of 20 ms at a sampling rate of 1000 Hz and record data at 60 Hz with a battery life of around 6–7 hours (Xsens MVN awinda, Xsens Technologies B.V., Enschede, Netherlands, and MVN Analyze software Version 2020.0.2.0). Additionally, based on the individual body dimensions (e.g., body height, knee, hip, and ankle heights, and shoe length), the IMU-data was fused into a biomechanical model and calibrated statically in a neutral standing pose and dynamically during walking for about 3 m and returning. Although the gait can be analyzed with a reduced sensor set of seven IMUs (placed on the sacrum, upper legs, lower legs, and on both feet), all the experiments were recorded with the whole-body set up to potentially be able to analyze additional gait-related aspects such as arm swing or counter rotation in the trunk. The kinematic data is collected for all 23 body segments and 22 adjacent joints and subsequently stored as htmx-files. It has been shown to be applicable to a flexible environment and demonstrated good usability for experienced and non-experienced professionals [20]. 

### 2.2. Cloud-Based Gait Analysis Report

Recent system advances allow for online gait analysis. The recorded movement files can be uploaded onto the Xsens motion cloud and analyzed online. Further, based on the event detection of heel strike and toe off, spatial and temporal gait parameters are calculated [19]. The gait parameters presented in the report include the number of foot contacts, temporal and spatial parameters, joint kinematics in three dimensions, center of mass, and limb segment accelerations. For the purpose of this study, the following gait metrics were included in the analysis of test-retest reliability, measurement error, and hypothesis testing per gait recording:Speed in meters per second was defined as the average walking pace of the gait recording.Steps were defined as the total number of steps identified in both legs based on heel strike detection.Step length in centimeters was defined as the distance between the heel strike position of the first foot and the heel strike of the opposite foot.Step width in centimeters was defined as the medial lateral distance between the heel strikes of the corresponding foot.Swing phase in seconds was defined as the time required from toe off to heel strike of one leg.Stance phase in seconds was defined as the time elapsed from heel strike to toe off in one leg.Hip flexion/extension in degrees were defined as the absolute range of motion from the minimum to the maximum hip joint angle in the sagittal plane.Knee flexion/extension in degrees were defined as the absolute range of motion from the minimum to the maximum knee joint angle in the sagittal plane.Ankle flexion/extension in degrees were defined as the absolute range of motion from the minimum to the maximum ankle joint angle in the sagittal plane.

### 2.3. Experimental Protocol

The participants were invited to attend one measurement appointment, consisting of two sessions of 1 to 1.5 h each, separated by at least 30 min. The study participants provided demographic and stroke-related information (e.g., age, sex, body height, body weight, side of stroke, time since stroke, ambulation category, and assistive devices) at the beginning of the experiments. We set up the wearable sensing suit by applying the sensors, including measures of participants’ individual, predefined body dimensions, and performing the static and dynamic calibration procedures. In addition, after successful calibration and online checking of motion replay in the software, three clinical ambulation tests were performed and recorded with the sensor suit, as illustrated in Figure 1. The sensor recordings were manually recorded with a button press, which we validated by visual inspection of the animated video or sensor data, and manually corrected if needed.

The recording for each session started with the execution of the Timed “Up and Go” Test (TUG). In this test, the tested person sits on a chair with armrests and is asked to stand up after the start signal, walk three meters, turn around, and return to the chair while the time needed to complete the task is measured [21,22]. The individuals requiring more than 30 s for test completion are likely to have severe mobility restrictions whereas those who perform the test in less than 10 s tend to be unrestricted walkers [23,24,25].

The 10 m Walk Test (10MWT) [26] was performed three times per session to assess the average walking speed and step length. After the start signal, the tested person is asked to walk as quickly and safely as possible beyond the 10 m mark. The examiner records the number of steps taken. The 10MWT is highly recommended in the stroke population [25], has normative data available [27], and shows excellent reliability [28]. 

Finally, the six-minute walk test (6MWT) was performed and recorded for assessing functional walking performance in persons with cardiopulmonary and metabolic disorders [29]. The tested person is asked to walk as many meters as possible on a predefined walkway for a time of 6 min. The distance in meters and any breaks needed are documented. The 6MWT is a widely used assessment tool with available age- and gender-specific norms for many countries [30]. In addition, before and after the exercise, the participants’ perceived effort and exertion were measured using the Borg Rating of Perceived Exertion (RPE) Scale [31]. The scale included RPE values from 6 to 20, with 6 being no exertion at all and 20 being maximal exertion. 

During the break, participants were asked to rest in a sitting position. The sensors were removed if necessary and recalibrated prior to the second session. The participants’ physical activity and strength were assessed during the break, ensuring that they had sufficient time to relax.

The International Physical Activity Questionnaire (IPAQ) is a 27-item, self-reported measure of physical activity [32]. The duration and frequency of different activities over the last 7 days are assessed for five domains: job, transportation, housework/family care, recreation/sport/leisure-time, and time spent sitting. The results are reported by means of the metabolic equivalent of task time (MET-minutes/week) and sitting hours/week. The IPAQ can be applied to a mixed population and has adequate to excellent test-retest reliability [32,33].

The strength in the affected leg was tested with the Motricity Index of the lower extremity (MI-LE) for dorsiflexion, knee extension, and hip flexion [34]. In addition, each movement was first performed dynamically throughout the complete joint range of motion and rated for isometric strength according to the MRC grades. The MI-LE shows excellent test-retest reliability [35] and criterion validity in comparison with a hand-held dynamometer [36].

### 2.4. Statistical Analysis

The descriptive statistics of the gait report parameters (speed, steps, step length, step width, swing phase, stance phase, hip flexion/extension, knee flexion/extension, ankle flexion/extension) for normal distribution were tested by use of the Shapiro-Wilk test and visual inspection of the QQ-plots; the results are presented as means and standard deviations. We statistically analyzed the test (first measurement) and retest (repetition of the measurements, after a break of at least 30 min) for reliability and validity. Test-retest reliability defines the extent to which scores for patients who have not changed are similar across repeated measurements [37]. The test-retest reliability of the gait parameters was investigated by the intraclass correlation coefficient (ICC 2,1) and the corresponding 95% confidence interval (CI) based on a single rater, absolute agreement 2-way random model of the test and retest data [38]. The ICCs for the 10MWT and 6MWT data were separately calculated to consider the potential effects of walking distance. The ICC values were interpreted as follows: poor, <0.5; moderate, 0.5–0.75; good, 0.75–0.9; excellent reliability, >0.9 [39]. The Bland-Altmann plots [40] were used to visualize the mean of differences between test and retest for the 10MWT and 6MWT data. In addition, the measurement error represents the systematic error and random error of a score that are not attributable to real changes [37]. It is determined by the standard error of measurement (SEM = SD × √1 − R). The SEM indicates the absolute reliability for individual subjects and different measurement timepoints and the associated minimal detectable change (MDC_95_ = SEM × 1.96 × √2) [41].

The hypothesis test of construct validity is defined as the degree to which the scores of an outcome are consistent with the hypothesis in relation to the scores of other instruments or differences between relevant groups [37]. The test and retest data from the 10MWT and 6MWT were used for this analysis. We applied Pearson correlation for normally distributed data and Spearman correlation in cases of non-normal distribution. The correlation scores (r) were obtained as follows: poor, r < 0.25; fair, 0.25 ≤ r < 0.5; moderate, r = 0.5–0.75; or high, r > 0.75 [42]. It was hypothesized that there are moderate-to-high positive correlations between muscle strength (MI-LE) and gait parameters of speed, step length, and range of motion in the hip, knee, and ankle. Furthermore, it was expected that there would be moderate-to-high positive correlations between these gait parameters and physical activity (IPAQ), physical functioning (6MWT), and clinically evaluated mean gait speed (10MWT). Further, it was hypothesized that there are moderate-to-high negative correlations between the time needed to perform the ambulation and balance tasks (TUG) and the named gait parameters. The scatterplots were used to illustrate relationships between strength, endurance, and balance/mobility and the gait analysis parameters by considering different walking ability subgroups (normal ambulator (>1.1 m/s), community ambulator (>0.8 m/s), limited community ambulator (0.4–0.8 m/s), and household ambulator (<0.4 m/s)) [43,44].

In order to determine the discriminability between the affected and less-affected leg for each gait parameter and potential effects between the test conditions, a two-way ANOVA was applied for each gait parameter. In the case of unmet assumptions for ANOVA testing (e.g., not normally distributed data or no homogeneity of variance), the Wilcoxon signed rank test was used to test for significant differences between the affected and less-affected sides. Finally, all statistical analysis was performed using R software (version 25.0, IBM Corp., Armonk, NY, USA).

## 3. Results

A total of 28 chronic stroke participants were included in the study; their demographic information is presented in Table 1. The subjects’ ages ranged from 44 to 90 years old, with 19 men and 9 women. The body height and BMI ranged from 149 to 187cm and 19.23 to 35.58 kg/m^2^, respectively. A total of eight participants had pronounced deficits in ambulation; eight were community ambulators; and 12 showed walking speeds comparable to the general age-matched population. In addition, nine participants used a walking stick for ambulation in daily life, and four participants used a foot orthosis. Further, all gait parameters for normal distribution were tested with the Shapiro-Wilk test and visually inspected in QQ-plots. The Shapiro-Wilk test did not confirm a normal distribution for one of the gait metrics. However, the inspection of the QQ-plots revealed a linear distribution of data points along the reference line for most of the investigated gait metrics.

### 3.1. System Usability

The inertial sensing suit was successfully operated by four naive users (one medical student, two physiotherapists, and one movement scientist) after one training session by an experienced user. The system setup, including at least acceptable calibration, was possible for all participants, with a mean time needed to set up and calibrate of 12 ± 5.5 min. The process of uploading and analyzing 10MWT data onto the cloud-based application was completed within seconds, while processing 6MWT data took around 3 min.

Several recordings could not be processed online in the gait analysis application. Only 6 of 112 TUG recordings (5.4%) resulted in gait reports, mainly because of the too-low number of steps during the TUG, which we excluded from the analysis. A total of 216 gait recordings were included in the analysis (6MWT, n = 56; 10MWT, n = 160). A few failures occurred during the recording. In two participants, a sensor dropped during ambulation testing. In one case, the sensor fell off the right upper leg; in the other case, the right foot sensor fell during the 6MWT at 5:10 min. One of the participants reported some discomfort due to the attached sensors. More specifically, the patient had the illusion that the sensor he was wearing on the affected upper leg was too loosely attached and falling down. 

### 3.2. Test-Retest Reliability

The ICC and SEM analyses included six trials of the 10MWT and two trials of the 6MWT per participant. A moderate-to-good test-retest reliability for the majority of gait metrics was observed in the 10MWT (ICC 0.46–0.88) and good-to-excellent reliability for most of the gait metrics in the 6MWT (ICC 0.63–0.99). The ICCs and the SEMs of all gait variables are shown in Table 2.

The minimal detectable change at the 95% confidence interval (MDC_95_) in step lengths was on average 20.64 cm for the 10MWT data and 5.28 cm for the 6MWT data; step width MDC_95_ was 5.03 and 3.81 cm for the 10MWT and 6MWT, respectively. The MDC_95_ of the swing phase and stance phase duration ranged from 0.15 to 0.03s and from 0.32 to 0.06s, respectively. Further, the MDC_95_ of the three joint angle ranges spanned from 8.45 to 16.99 degrees in the 10MWT data and from 5.79 to 19.93 degrees in the 6MWT data.

The Bland-Altman plots presented in Figure 2a,b, illustrate the mean of the difference against the test and retest means by considering the limits of agreement in terms of the 95% CI. Figure 2a,b show the Bland Altman plots per gait parameter during the 10MWT and 6MWT, with smaller margins in Figure 2b for the 6MWT data than in Figure 2a for the 10MWT, indicating a lower measurement error. 

The speed in meters per second measured during 10MWT resulted in a slightly increased measurement mean, as indicated in Figure 2a, and lower limits of agreement of the differences between test and retest in the 6MWT data, as shown in Figure 2b. The means of measurements of step length, step width, swing phase, and stance phase, as well as hip, knee, and ankle flexion/extension were comparable for the 10MWT and 6MWT recordings. In addition to the similar means of measurement of these gait metrics, the limits of agreement were narrower, with more data points falling within these limits in the 6MWT recordings than in the 10MWT recordings.

### 3.3. Hypothesis Testing Reliability

The results of hypothesis testing revealed a moderate-to-high correlation between most clinical and kinematic gait measures, except for step widths and the IPAQ and the other measures, as shown in Figure 3. The steps correlated with the TUG, speed, and distance of the 6MWT for the 10MWT data and the 6MWT data separately. As expected, the distance of the 6MWT strongly correlated with step length (r = 0.84, *p* = 0.003) and moderately with stance phase (r = −0.74, *p* < 0.001), hip flexion/extension (r = 0.67, *p* < 0.001), and knee flexion/extension (r = 0.57, *p* < 0.001). Similarly, high positive correlations were observed between the mean speed of the 10MWT and speed (r = 0.9, *p* < 0.001), step length (r = 0.83, *p* < 0.001), and stance phase (r = −0.80, *p* < 0.001), and moderate correlations with hip flexion/extension (r = 0.74, *p* < 0.001) and knee flexion/extension (r = 0.56, *p* < 0.001). 

The TUG was highly correlated with speed (r = −0.84, *p* < 0.001), step length (r = −0.85, *p* < 0.001) and stance phase (r = 0.81, *p* < 0.001), but moderately with hip flexion/extension (r = −0.54, *p* < 0.001). Finally, the MI-LE moderately correlated with speed (r = 0.54, *p* = 0.003) and stance phase (r = −0.54, *p* < 0.001).

Figure 4 shows the relationship between step length, stance phase, hip flexion/extension, and knee flexion/extension with the clinical measures 6MWT, TUG, and MI-LE by considering different ambulation subgroups. The scatter plots indicate almost linear relationships between step length and stance phase of the affected leg and the TUG and 6MWT, as well as between the hip and knee joint ranges of the affected leg and the TUG and 6MWT. 

A clear linear and hierarchical order of data was observed in terms of smaller step length, longer stance phase duration, and decreased absolute hip and knee joint range of motion in household ambulators compared with community and normal ambulators, and moderate-to-strong correlations in terms of decreased walking distance in the 6MWT and increased time needed to complete the TUG. The relationships between leg strength, as measured by the MI-LE, and step length, stance duration, and hip and knee joint range of motion were less conclusive.

### 3.4. Discriminability between Affected and Less-Affected Leg and 10MWT and 6MWT

Except for step length and step width, all gait parameters were significantly different on the affected and less-affected sides, as shown in Table 3 and Figure 5. Step length was not statistically significantly different between the affected and less-affected lower limbs (F(1) = 0.933, *p* = 0.335) and the test condition (F(1) = 0.711, *p* = 0.400). The stride width was not significantly different between the lower limb sides (F(1) = 0.014, *p* = 0.9045), but was between test conditions (F(1) = 6.646, *p* = 0.010).

The swing phase and stance phase did not confirm the homogeneity of variance, as tested by the Levene test (F(3) = 8.371, *p* < 0.001, and F(3) = 1.499, *p* = 0.017). The two-way ANOVA results revealed a significant difference in swing phase by side (F(1) = 44.355, *p* < 0.001), which was confirmed by the results of the Wilcoxon signed rank test (V = 16989, *p* < 0.001) and the lack of difference between the 10MWT and the 6MWT data (F(1) = 2.359, *p* = 0.125). The stance phase showed statistically-significant differences by side (F(1) = 7.385, *p* = 0.007) that were confirmed by non-parametric testing (V = 3130.5, *p* < 0.001) and the lack of differences between the test conditions (F(1) = 2.254, *p* = 0.134). 

The hip flexion/extension significantly differed between the affected and less-affected sides (F(1) = 29.94, *p* < 0.001) and between the 10MWT and 6MWT (F(1) = 21.79, *p* < 0.001). The knee and ankle flexion/extension were similarly significantly different for the factors side and test, whereas knee flexion/extension did not confirm the test for homogeneity (Levene’s test F(3) = 6.385, *p* < 0.001).

The range of motion in knee flexion/extension was significantly lower in the affected than in the less-affected lower limb (F(1) = 39.60, *p* < 0.001), confirmed by the results of non-parametric testing (V = 4894, *p* < 0.001) and lower in the 10MWT than in the 6MWT (F(1) = 17.73, *p* < 0.001). The ankle flexion/extension significantly differed by side (F(1) = 21.5, *p* < 0.001) and test condition (F(1) = 45.1, *p* < 0.001).

## 4. Discussion

The objective of this study was to determine the usability, reliability, and validity of a sensor-based gait analysis system in patients with stroke-related gait impairments, ranging from household mobility with a walking speed <0.4 m/s to persons with a normal walking speed of >1.1 m/s. The sensor suit could be operated by health professional students after one training session. The gait recordings of 10-m or more of walking could be collected and analyzed within a total time of approximately one hour, depending on the data file size. In addition, all tested gait parameters except for the swing phase in the less-affected side during the 10MWT had good-to-excellent test-retest reliability, with overall excellent reliability and reduced measurement errors in the 6MWT. These results are in agreement with those obtained in studies in healthy populations [19,45]. The validity of the gait parameters in terms of hypothesis testing demonstrated moderate-to-high correlations between the main clinical and kinematic gait measures. The associations between larger step length and range of motion were confirmed in the hip and knee flexion/extension on the one side and longer distances in the 6MWT, faster walking speed in the 10MWT, and less time needed in the TUG; and moderate relationships with increases in the strengths of the affected lower limb, as measured with the MI-LE, on the other side. These results are in line with existing findings [46,47,48,49] and support the idea that kinematic gait parameters reflect similar and relevant measurement constructs as clinical scales to assess lower limb function after stroke. Finally, differences were observed in the gait metrics between the affected and less-affected legs, as well as between the 10MWT and 6MWT, with a shorter stance phase, a prolonged swing phase, and a decreased range of motion in the hip, knee, and ankle joints of the affected leg compared with the less-affected leg. During the 6MWT, all joint ranges were larger and step width was shorter when compared with the metrics of the 10MWT, which is in accordance with existing research on different walking speeds [4,50] and distances [51]. The 6MWT was performed on a 60 m long pathway, which provided reasonable space for accelerating and decelerating walking speed and harmonizing gait patterns in contrast to the 10MWT set up.

The results of test-retest reliability are in line with those of other studies on the reliability of sensor-based gait analysis [45,52]. The measurement error of step length and width (range: 1.37–7.94 cm) found in this study is comparable to the results obtained from the performance analysis in 35 healthy subjects (root mean square error, RMSE, range: 2.61–6.82 cm) of the sensor-based gait report system [19]. However, in comparison with a camera-based motion capture system, the sensor-based gait analysis system showed errors in step length (−1.43 ± 2.26 cm), step width (−4.48 ± 5.02 cm) and the temporal parameters of double support time (−0.19 ± 0.04 s); hence, these measures tend to be systematically underestimated [19]. The accuracy of the sensor-based joint angle estimation of the system was proven to be excellent during walking at different speeds [53] as well as during stair climbing and sit-to-stand activities [45]. In comparison to camera-based gait analysis, data on joint angle estimations of the hip, knee, and ankle acquired with wearable sensors during walking indicated high intrasubject but low intersubject accuracy [14,54,55]. The finding of improved reliability in longer-lasting gait recordings containing more steps and gait cycles is consistent with the findings in the literature. Hansen et al. investigated the reliability of an IMU-based gait parameter of 4 and 20 m in patients with neurological diseases. It was observed that there is poor reliability for most of the temporal gait metrics and good reliability for the number of steps, with overall increases in reliability with walking distance. Moreover, in contrast to the presented sensor setup, only one IMU at the pelvis was used for the gait analysis [56]. Furthermore, the effect of movement velocity needs to be considered in sensor-based gait analysis because inaccuracies were reported at low walking speeds of 0.4 m/s, probably due to the speed dependency of sensor-based metrics [57]. In line with these findings, algorithms for gait detection were reported to perform less accurately depending on the severity of deviations from normal human gait due to diseases such as stroke [58]. In summary, the results indicate that gait recordings of more than 10 m or at least 15 steps [59], enable a reliable gait analysis of movement characteristics in persons after stroke. 

The correlation between clinical and sensor-based gait measures of the affected leg followed the expected assumptions for most of the outcomes. An increase in step length showed strong relationships with improved results in the 10MWT, 6MWT, and TUG, as well as moderate associations with MI-LE [51]. Further, an increase in stance phase duration was negatively associated with the results on the 10MWT, 6MWT, MI-LE, and TUG. The increases in the range of motion in hip and knee flexion/extension showed moderately positive associations with improvements in the 10MWT, 6MWT, and TUG. Additionally, both hip and knee joint ranges strongly correlate with each other, underpinning the synergistic relationship of both joints during walking [47]. The increases in swing phase duration were moderately associated with worse results in 6MWT, 10MWT, and TUG, as well as moderately associated with lower knee flexion/extension ranges. There was a prolonged swing phase duration that was reported in stroke-related gait impairments [60] and might be caused by muscle weakness and/or spasticity in the affected limb as well as step length differences [51,60,61]. Furthermore, Garland et al. found relationships between increased swing phase duration and hamstring muscle weakness [62]. This study did not reveal any associations between physical activity, as assessed by the IPAQ, and the gait parameters. The IPAQ results showed only a moderate correlation between the 6MWT and TUG. The inconclusive results between the IPAQ and gait metrics may be explained by the different measurement domains assessed and/or the poor validity reported between accelerometer monitoring and the IPAQ [33,63]. 

The stroke-specific differences between the affected and less-affected legs, such as the diminished stance phase duration, longer swing phase duration, and decreased joint range of motion in the affected leg [50], fit well into the stroke-specific gait characteristics. The results on stance duration of the less-affected leg are in line with the findings on elderly healthy subjects reporting a mean stance duration of 0.71s [64]. The decreased range of motion in the affected ankle joint is explained by the effect of foot orthoses being used in four study participants. The foot orthoses are known to limit the range of motion in the ankle while preventing foot drop, instability in the direction of supination, easing foot clearance, and thereby increasing walking speed and other gait kinematics [65]. Further, diminished ankle joint ranges were explained by frequently observed weaknesses of dorsiflexion as well as plantar flexion at terminal stance and toe off [66]. There were no significant associations between strengths in dorsiflexion, as measured with the MI-LE, and ankle joint range of motion in flexion/extension. However, these results may have differed if we had recorded ankle joint ranges without orthoses substitution. 

In summary, these findings underpin the potential value of joint kinematics to further characterize stroke-specific movement disorders and justify sensor setups with several IMUs attached to more body segments for predefined periods. However, the setup, which includes at least seven IMUs on the lower body, is a limitation to the tolerance of long-term recordings. 

The limitations that need to be considered are the small (n = 28) and unbalanced sample size with four participants representing limited household walkers. In addition, studies on a larger population, allowing for a systematic exploration of the effects of age, sex, assistive devices, and gait pathologies on different severity levels, are recommended to ensure a balanced dataset and the formation of representative subgroups. Further, the most common gait parameters were the focus, even though we could have evaluated several other spatial and temporal parameters provided in the report. The validation of gait events such as heel strike contact and toe off were outside the scope of this study, as were other available metrics such as the foot progression angle, single or double support, gait cycle phases (e.g., loading response, midstance, terminal stance, and pre-swing), pelvis orientation, center of mass tracking, and upper and lower leg acceleration.

Furthermore, future research will likely increase the accuracy and robustness of sensor-based human motion analysis. In parallel, developments of nanogenerator-based wearable devices [67] and smart textiles, such as embedded bending sensors for measuring joint movements [68], constitute potential future applications. Wearable sensors may profoundly contribute to treatment and assessment opportunities in different real-world settings, such as rehabilitation clinics or a person’s home, allowing the exploration of the impact of environmental factors on gait performance [69]. Such systems provide complex and accurate data on movement quality in a more time-efficient and less-restricted manner than classical movement laboratories. However, the systematic errors of sensor-based systems and the impact of disease-specific movement characteristics on measurement accuracy need further elaboration.

Finally, although sensor setups on the foot and pelvis are less burdensome to wear and allow continuous real-life data recordings, the information on joint angles in the main degrees of freedom of the lower limb seems to be relevant for the characterization of stroke-related gait impairments and should be considered in future research.

## 5. Conclusions

The proposed gait analysis system demonstrated user-friendly applicability and assessment quality in persons with stroke-related gait disorders. A walking period of at least 10m and at least 15 steps are suggested to assure a reliable gait analysis. In addition, gait measures such as step length, stance phase, and hip flexion/extension range moderately to strongly relate to clinical gait measures. This study supports the idea that aspects of movement quality during walking can be objectively captured and processed with relatively low technological and environmental effort. Finally, the associated knowledge on movement characteristics can be helpful for accurate and time-efficient assessments in different environmental surroundings, presenting a potential tool for monitoring the movement quality in supervised and unsupervised training situations.

## Figures and Tables

**Figure 1 sensors-23-00624-f001:**
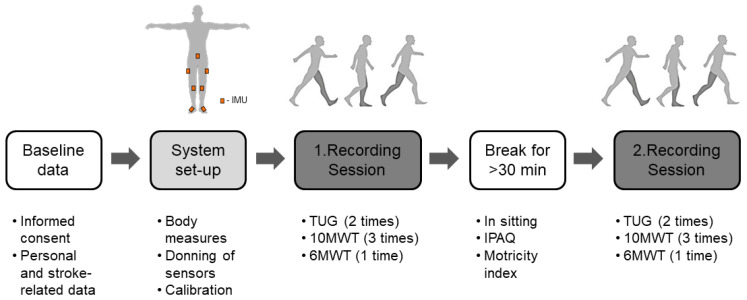
Experimental protocol. The timed up and go test (TUG), ten meter walk test (10MWT), and six-minute walk test (6MWT) were recorded with an IMU-based sensor suit in sessions 1 (test) and 2 (retest).

**Figure 2 sensors-23-00624-f002:**
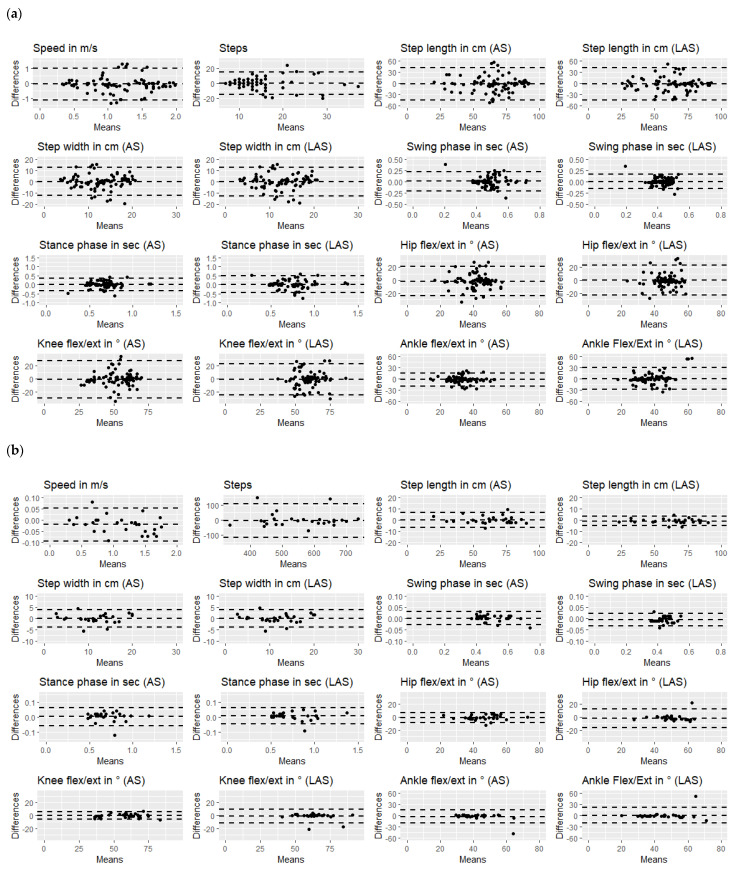
Bland Altman plots per gait metric during (**a**) 10MWT and (**b**) 6MWT.

**Figure 3 sensors-23-00624-f003:**
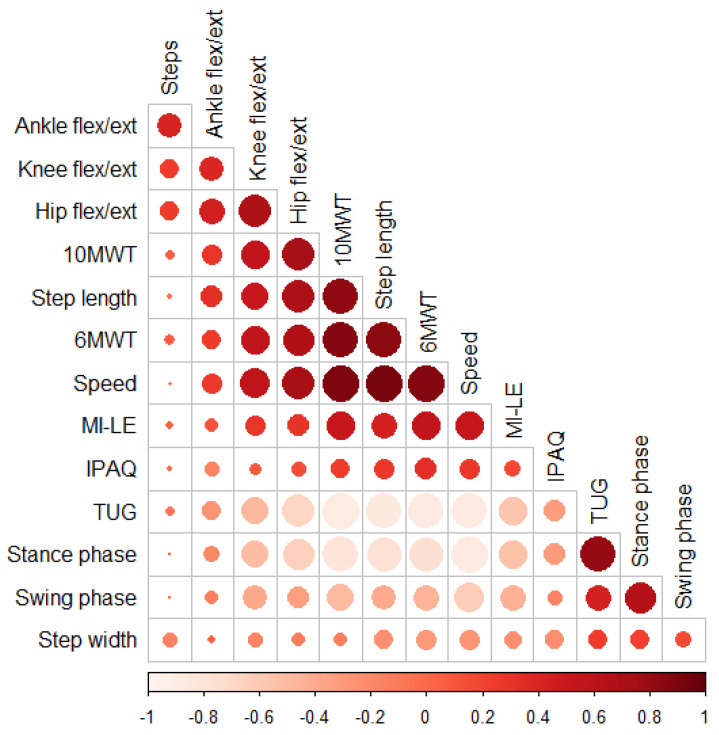
Correlation matrix of clinical and kinematic gait measures.

**Figure 4 sensors-23-00624-f004:**
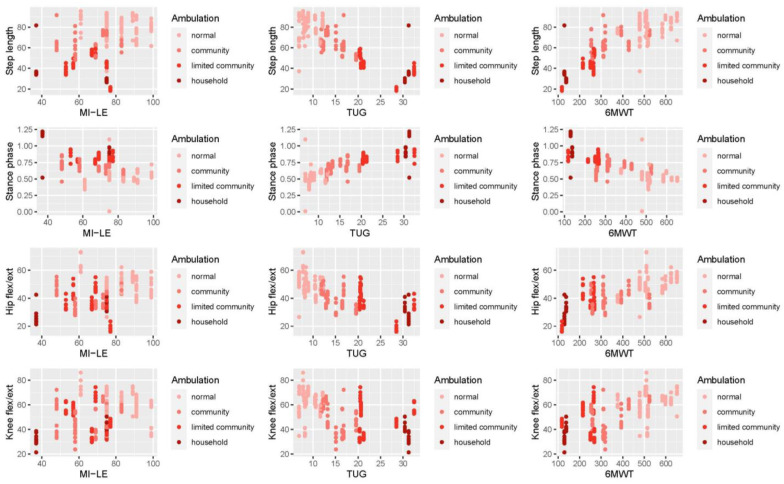
Relationships between clinical, kinematic gait metrics (step length, stance phase, and hip and knee flexion/extension) and ambulatory deficits (normal, community, and limited community or household ambulation). MI-LE, motricity index of the lower extremity; TUG, timed up and go test; 6MWT, six minute walk test.

**Figure 5 sensors-23-00624-f005:**
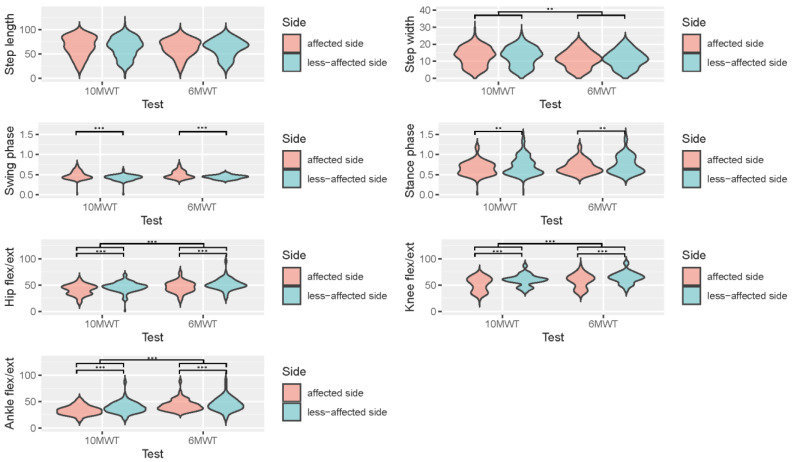
Gait metrics (average step length, step width, swing phase, stance phase duration, and absolute joint ranges of hip, knee, and ankle flexion/extension) differences between legs and between 10MWT and 6MWT. ** indicates statistical significance of *p* < 0.01 and *** indicates statistical significances of *p* < 0.01.

**Table 1 sensors-23-00624-t001:** Participant characteristics.

Characteristic	N = 28
Sex, female/male	9/19
Age in years, mean (SD)	62.04 (11.68)
Body height in cm, mean (SD)	172.3 (9.89)
Body mass index in kg/m^2^, mean (SD)	25.76 (3.30)
Paretic body side, left/right	15/13
Months since stroke, mean (SD)	63.71 (51.85)
Initial stroke severity NIHSS, median (Q1-Q3), N = 23	8.5 (6–10)
Household ambulators (<0.4 m/s), n (%)	2 (7.14)
Limited community ambulators (0.4–0.8 m/s), n (%)	6 (21.43)
Community ambulators (>0.8 m/s), n (%)	8 (28.57)
Normal ambulators (>1.1 m/s), n (%)	12 (42.86)
Assistive device, n (%)	9 (32)
Foot orthoses, n (%)	4 (14)
MI-LE Total, median (Q1–Q3)	75 (60–83)/100
TUG in seconds, mean (SD)	15.42 (7.76)
10MWT mean speed in m/s, mean (SD)	1.03 (0.45)
6MWT in meters, mean (SD)	384.3 (156.4)
IPAQ in MET/week, mean (SD)	2493 (2014)

Legend: MET—metabolic equivalent of task; Q1–Q3—first to third quartile; SD—standard deviation.

**Table 2 sensors-23-00624-t002:** Test-retest reliability and measurement error per gait metric and test condition.

Gait Metric	10MWT			6MWT		
	Mean SD	ICC (95% CI)	SEM	Mean SD	ICC (95% CI)	SEM
Speed (m/s)	1.19 ± 0.52	0.84 (0.73–0.92)	0.21	1.11 ± 0.45	0.99 (0.99–0.99)	0.03
Steps	14 ± 7.13	0.82 (0.71–0.91)	3.02	554.4 ± 105.08	0.86 (0.73–0.93)	38.92
Step length, cm (AS)	65.39 ± 20.19	0.88 (0.80–0.94)	6.95	62.57 ± 17.40	0.98 (0.96–0.99)	2.25
Step length, cm (LAS)	63.05 ± 20.26	0.87 (0.81–0.92)	7.94	62.26 ± 17.67	0.99 (0.98–0.99)	1.56
Step width, cm (AS)	12.89 ± 5.44	0.80 (0.71–0.87)	1.80	11.34 ± 4.91	0.92 (0.84–0.96)	1.37
Step width, cm (LAS)	12.81 ± 5.49	0.80 (0.71–0.87)	1.83	11.34 ± 4.91	0.92 (0.84–0.96)	1.38
Swing phase, s (AS)	0.48 ± 0.10	0.60 (0.43–0.78)	0.06	0.49 ± 0.09	0.99 (0.97–0.99)	0.01
Swing phase, s (LAS)	0.43 ± 0.07	0.46 (0.28–0.67)	0.05	0.44 ± 0.05	0.95 (0.90–0.98)	0.01
Stance phase, s (AS)	0.66 ± 0.18	0.67 (0.50–0.82)	0.10	0.69 ± 0.17	0.98 (0.97–0.99)	0.02
Stance phase, s (LAS)	0.71 ± 0.23	0.64 (0.47–0.80)	0.13	0.75 ± 0.22	0.99 (0.98–0.99)	0.02
Hip flex/ext, ° (AS)	41.28 ± 9.96	0.87 (0.79–0.94)	3.53	46.31 ± 10.34	0.93 (0.86–0.97)	2.73
Hip flex/ext, ° (LAS)	46.46 ± 9.33	0.73 (0.58–0.86)	4.85	51.55 ± 10.72	0.76 (0.54–0.88)	5.28
Knee flex/ext, ° (AS)	51.76 ± 13.22	0.88 (0.80–0.94)	4.54	58.10 ± 12.84	0.97 (0.94–0.99)	2.09
Knee flex/ext, ° (LAS)	59.44 ± 10.72	0.84 (0.74–0.92)	4.25	64.21 ± 11.03	0.89 (0.78–0.95)	3.69
Ankle flex/ext, ° (AS)	33.21 ± 8.78	0.88 (0.79–0.94)	3.05	42.71 ± 10.77	0.63 (0.35–0.81)	6.56
Ankle flex/ext, ° (LAS)	38.81 ± 11.20	0.70 (0.54–0.84)	6.13	44.86 ± 12.74	0.68 (0.42–0.84)	7.19

Legend: AS, affected side; CI, confidence interval; LAS, less-affected side; ICC, intraclass correlation coefficient; SEM, standard error of measurement.

**Table 3 sensors-23-00624-t003:** Comparison of the affected vs. less-affected leg and 10MWT vs. 6MWT per gait metric.

Gait Metric	10MWT		6MWT		Levene	ANOVA	
	Affected Leg	Less-Affected Leg	Affected Leg	Less-Affected Leg		Leg	Test
	Mean SD	Mean SD	Mean SD	Mean SD	*p*-Value	*p*-Value	*p*-Value
Step length, cm	65.39 ± 20.19	63.05 ± 20.26	62.57 ± 17.4	62.26 ± 17.67	0.178	0.335	0.400
Step width, cm	12.89 ± 5.44	12.81 ± 5.49	11.34 ± 4.91	11.34 ± 4.91	0.452	0.905	**0.010**
Swing phase, s	0.48 ± 0.1	0.43 ± 0.07	0.49 ± 0.09	0.44 ± 0.05	**<0.001**	**<0.001**	0.126
Stance phase, s	0.66 ± 0.18	0.71 ± 0.22	0.69 ± 0.17	0.74 ± 0.22	**0.017**	**0.007**	0.135
Hip flex/ext, °	41.28 ± 9.96	46.46 ± 9.33	46.31 ± 10.34	51.55 ± 10.72	0.214	**<0.001**	**<0.001**
Knee flex/ext, °	51.76 ± 13.22	59.44 ± 10.72	58.1 ± 12.84	64.21 ± 11.03	**<0.001**	**<0.001**	**<0.001**
Ankle flex/ext, °	33.21 ± 8.78	38.81 ± 11.2	42.71 ± 10.77	44.86 ± 12.74	0.157	**<0.001**	**<0.001**

Statistical significance is indicated in bold.

## Data Availability

The data presented in this study is available on request from the corresponding author. The data are not publicly available due to privacy issues with individual gait recordings.

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
