# Peer review of "Reliability and Validity of a Wearable Sensing System and Online Gait Analysis Report in Persons after Stroke"

_sensors, 2023, doi:10.3390/s23020624_

Round 1
Reviewer 1 Report
This manuscript reports wearable sensing system for gait analysis. The manuscript can be accpet after the following issues are solved.
1, The system of the sensing is not solid, the author needs to cite more wearable sensors to support the background of the motion sensors, such as Nano Energy, 103 (2022) 107766.
2, What is the noverlty of the manucript, is the system bought from company or design by the authors?
3, The data in Figure 2 seems points, however, from the sensors the data should be curves, why?
4, Any similar research on this topic, the author can compare of them.
Author Response
Reviewer 1
This manuscript reports wearable sensing system for gait analysis. The manuscript can be accpet after the following issues are solved.
Thanks for the critical appraisal of our submitted manuscript. We carefully considered the reviewers comments to improve the quality of the article. Additionally, the revised manuscript is submitted for English language editing.
1, The system of the sensing is not solid, the author needs to cite more wearable sensors to support the background of the motion sensors, such as Nano Energy, 103 (2022) 107766.
Thanks for the critical comment and recommendation. We have checked the reference and included the following information into the discussion of the manuscript (lines 492-494): “In parallel, developments of nanogenerator-based wearable devices [67] and smart textiles, such as embedded bending sensors for measuring joint movements [68] constitute potential future applications.“
2, What is the noverlty of the manucript, is the system bought from company or design by the authors?
We acknowledge that the investigated system is a commercially available system and very well studied system for whole body human motion analysis. The novelty investigated in the current manuscript, is an online tool to generate gait reports based on sensor-based recordings. From a clinical perspective, such technologies should be operable within reasonable time and without extensive training needed, while assuring reliable and valid outcomes when being applied in a certain population such as people post-stroke. The submitted manuscript therefor might be focused rather on aspects important for clinical application than on novelty of the system.
3, The data in Figure 2 seems points, however, from the sensors the data should be curves, why?
We investigated the gait parameters, as defined in section 2.2 based on summary statistics per gait recording, such as the average speed or the absolute range of joint motions, and did not consider continuous data or joint angle curve that would be available in the automatized gait reports. For investigating test-retest reliability, we have decided to visualize the test-retest data in Bland Altman Plots.
4, Any similar research on this topic, the author can compare of them.
We have focused the discussion on comparisons to other IMU-based gait analysis, such as Hansen et al 2022, who similarly investigated the reliability of IMU-based gait analysis in elderlies. The state of art on the relationship between sensor-based and camera-based motion analysis was additionally discussed. Further comparisons between different wearables lay outside the scope of the presented study and were therefor only briefly indicated in the section of future directions.
Reviewer 2 Report
In this paper, the authors investigate the reliability and validity of a sensor-based gait analysis system in stroke survivors with different severities of gait deficits. For this purpose, 28 chronic stroke survivors (nine females, age: 62.04±11.68 years) with mild to moderate walking impairments performed a set of ambulatory assessments (3x 10MWT, 1x 6MWT per session) two times while being equipped with a sensor suit. The results indicate all gait parameters showed moderate to strong correlations with clinical scales reflecting lower limb function. The confirmation of reliability and validity for the sensor-based gait parameters support the applicability in individuals with stroke-related walking impairments. This article is clear, concise, and suitable for the scope of the journal. Several small suggestions are supplied:
1. Suggest the authors supply more detail in sentences about the Bland Altman Plots per gait metric during 10MWT and 6MWT.
2. Suggest the authors supply more detail in sentences about the relationship between clinical, kinematic gait metrics and ambulatory deficits.
3. Suggest the authors introduce other sensor technology for stokes rehabilitation, such as:
Embedded FBG-Based Sensor for Joint Movement Monitoring, IEEE Sensors Journal 21(23):26793-26798, 2021
Author Response
Reviewer 2
In this paper, the authors investigate the reliability and validity of a sensor-based gait analysis system in stroke survivors with different severities of gait deficits. For this purpose, 28 chronic stroke survivors (nine females, age: 62.04±11.68 years) with mild to moderate walking impairments performed a set of ambulatory assessments (3x 10MWT, 1x 6MWT per session) two times while being equipped with a sensor suit. The results indicate all gait parameters showed moderate to strong correlations with clinical scales reflecting lower limb function. The confirmation of reliability and validity for the sensor-based gait parameters support the applicability in individuals with stroke-related walking impairments. This article is clear, concise, and suitable for the scope of the journal. Several small suggestions are supplied:
Thank you very much for the positive review and the valuable suggestions.
Suggest the authors supply more detail in sentences about the Bland Altman Plots per gait metric during 10MWT and 6MWT.
Thanks for the kind suggestion. The following section has been included in lines 290-298: “It can be seen that speed in m/s measured during 10MWT resulted in slightly in increased mean of measurements as indicated in Figure 2(a) and lower limits of agreement of the differences between test and retest in the data of the 6MWT in Figure 2(b). The mean of measurements of step length, step width, swing phase, stance phase, as well as hip, knee and ankle flexion/extension are comparable for the 10MWT and the 6MWT recordings. Apart from the similar means of these gait metrics, the limits of agreement were narrower with more data points falling within these limits in the 6MWT recordings when compared to the 10MWT recordings.”
Suggest the authors supply more detail in sentences about the relationship between clinical, kinematic gait metrics and ambulatory deficits.
We acknowledge the lack of detail provided and included the following section: “Furthermore, the relationship between step length, stance phase, hip flex-ion/extension and knee flexion/extension with the clinical measures 6MWT, TUG and MI-LE by considering different ambulation subgroups, are shown in Figure 4. The scatter plots including the ambulation subgroups indicate almost linear and hierarchi-cally ordered relationships between step length, stance phase of the affected leg and the TUG and 6MWT as well as between the hip and knee joint ranges of the affected leg and the TUG and 6MWT.
A clear linear and hierarchical order of data can be seen in terms of smaller step length, longer stance phase duration, as well as decreased absolute hip and knee joint range of motion in household ambulators when compared to community and normal ambulators and moderate to strong correlations in terms of decreased walking distance in 6MWT and increased time needed to complete the TUG. The relationship between leg strength as measured by the MI-LE and step length, stance duration as well as hip and knee joint range of motion were less conclusive.”
Suggest the authors introduce other sensor technology for stokes rehabilitation, such as:
Embedded FBG-Based Sensor for Joint Movement Monitoring, IEEE Sensors Journal 21(23):26793-26798, 2021
Thank you for the recommendation. We have included the reference in the following section (lines 492-494): “In parallel, developments of nanogenerator-based wearable devices [67] and smart textiles, such as embedded bending sensors for measuring joint movements [68] constitute potential future applications.“
Reviewer 3 Report
The paper covers very interesting information on patient rehabilitation based on new IMU systems.
The paper is mostly clear and understandable, however, some aspects could be more precise.
The overall merit was treated in a usually medical statistical way, it could be treated another way for more research clearance. Even though the data will be less compared with other research, it would be less biased (see comments below).
The discussion is well written, and the conclusions are as well.
Detailed remarks
L 95 – how was it checked?
L 126 – average speed? And others? For all reported tests and steps?
L 152 –the photo of the sensor suit would be useful
L155-175 – a better division of tests would be more useful
L 221-226 – the hypothesis should be in the introduction
L 196 – statistical analysis did not take into account the effects that were identified in L 146, which could be calculated another way by using analysis of variance for data of different distribution. If not the aspect of different subjects' characteristics should be mentioned in the discussion part.
The distribution of the data was not presented. It is mentioned in methods, some info is given in results as results from the specific analysis, but the distribution should be given in the first part of the results.
Figure 2 results are not visible enough. Perhaps two figures -a i b would be better. One page of small images is difficult to read. Even some text between could help and images would be greater.
Figure 4 –what is AS?
Figures 5 and 6 could be described in more detail in the legend. They all should stay alone.
Associations from L365 are not clear. Please check the text.
the results from intra/inter-person variation could be interesting.
L 445 – it would be useful to discuss the possible effects (eg. sex, age) here.
Very clear conclusions.
Author Response
Reviewer 3
The paper covers very interesting information on patient rehabilitation based on new IMU systems.
The paper is mostly clear and understandable, however, some aspects could be more precise.
The overall merit was treated in a usually medical statistical way, it could be treated another way for more research clearance. Even though the data will be less compared with other research, it would be less biased (see comments below).
We appreciate the reviewers evaluation of the submitted manuscript.
The discussion is well written, and the conclusions are as well.
Thank you very much.
Detailed remarks
L 95 – how was it checked?
Each participant was asked about any pre-existing gait impairments that were unrelated to the stroke during telephone recruitment. Furthermore, no other gait limitation became apparent during the experiments.
L 126 – average speed? And others? For all reported tests and steps?
Thank you for the questions. Indeed, we considered average speed, the total number of steps, as well as the average of step length, step width, swing phase, stance phase per gait recording. We have adapted the section as follows: “For the purpose of this study, the following gait metrics were included in the analysis of test-retest reliability, measurement error and hypothesis testing per gait recording”.
L 152 –the photo of the sensor suit would be useful
Thanks for the suggestion. Unfortunately, we do not have a photo of the sensor suit. For illustration of the sensor setup, we have included a setup schematic in Figure 1.
L155-175 – a better division of tests would be more useful
Sorry for the lack of clarity. The section has been reformulated, as follows: “Each recording session started with the execution of the Timed “Up and Go” Test (TUG). The tested person sits on a chair with armrests and is asked to stand up after the start signal, walk three meters, turn around and return to the chair, while the time needed to complete the task is measured. [21, 22]. Individuals requiring more than 30 seconds for test completion are likely to have severe mobility restrictions, whereas those who perform the test in less than 10 seconds tend to be unrestricted walkers [23–25].
The 10 Meter Walk Test (10MWT) [26] was performed three times per session to assess the average walking speed and step length. After the start signal, the tested person is asked to walk as quickly and safely as possible beyond the 10 m mark. The examiner takes the time and counts the steps taken. The 10MWT has been highly recommended in the stroke population [25], has normative data available [27] and has attested excellent reliability [28].
Finally, the Six-Minute Walk Test (6MWT) is performed and recorded for assessing functional walking performance in persons with cardiopulmonary and metabolic dis-orders [29]. The tested person is asked to walk as many meters as possible on a predefined walkway for the time of 6 minutes. The distance in meters and any breaks needed are documented. The 6MWT is a widely used assessment with available age- and gender-specific norms for many countries [30]. Before and after the exercise, the participants’ perceived effort and exertion was measured using the Borg Rating of Perceived Exertion (RPE) Scale [31]. The scale included RPE values from 6 to 20, with 6 being no exertion at all and 20 being maximal exertion.”
L 221-226 – the hypothesis should be in the introduction
Thanks for the recommendation. We have adapted the section in the introduction (line 83): “The third aim was to investigate the construct validity by means of hypothesis testing between the gait report kinematics and clinical scales on strength, mobility and physical activity in person after stroke.”
We decided to keep the specific hypothesis in the methods section, close to the experimental and outcome measure descriptions.
L 196 – statistical analysis did not take into account the effects that were identified in L 146, which could be calculated another way by using analysis of variance for data of different distribution. If not the aspect of different subjects' characteristics should be mentioned in the discussion part.
We acknowledge the reviewers criticism of not reporting statistical analysis in section 2.5 on the participant demographics, such as age, gender, body height, weight or assistive devices being used. We initially considered the factors of age, gender and BMI in the correlation analysis and in the linear mixed model analysis. None of these factors resulted in a meaningful significance in any of the analysis performed. As suggested, we have included a discussion of these aspects in the studies limitations: “Research on a larger population, allowing to systematically explore the effect of age, gender, assistive devices and gait pathologies on different severity levels is recommended to enable a balanced dataset and the formation representative subgroups.”
The distribution of the data was not presented. It is mentioned in methods, some info is given in results as results from the specific analysis, but the distribution should be given in the first part of the results.
We apologize for the lack of reporting and included the following information, as suggested, in lines 249-254: “All included gait parameter were tested for normal distribution by Shapiro-Wilk test and visually inspected in QQ-plots. The Shapiro-Wilk test did not confirm normal distribution for one of the gait metrics. However, the inspection of QQ-plots revealed linear distribution of data points along the reference line for most of the investigated gait metrics.”
Figure 2 results are not visible enough. Perhaps two figures -a i b would be better. One page of small images is difficult to read. Even some text between could help and images would be greater.
Figure 2 a and b were formatted to improve the readability. The following information were included in the text in lines 290-298: “It can be seen that speed in m/s measured during 10MWT resulted in slightly in increased mean of measurements as indicated in Figure 2(a) and lower limits of agreement of the differences between test and retest in the data of the 6MWT in Figure 2(b). The mean of measurements of step length, step width, swing phase, stance phase, as well as hip, knee and ankle flexion/extension are comparable for the 10MWT and the 6MWT recordings. Apart from the similar means of these gait metrics, the limits of agreement were narrower with more data points falling within these limits in the 6MWT recordings when compared to the 10MWT recordings.”
Figure 4 –what is AS?
AS was used for the affected leg, but mistakenly included. The information has been removed.
Figures 5 and 6 could be described in more detail in the legend. They all should stay alone.
The following text has been included in the legend caption (lines 325-328): “Relation between clinical, kinematic gait metrics (step length, stance phase, hip and knee flexion/extension) and ambulatory deficits (normal, community, limited community or household ambulation). MI-LE, motricity index of the lower extremity; TUG, timed up and go test; 6MWT, six-minute walk test.”
Lines 350-352: “Gait metric (average step length, step width, swing phase, stance phase duration and absolute joint ranges of hip, knee and ankle flexion/extension) differences between legs and between 10MWT and 6MWT.”
Associations from L365 are not clear. Please check the text.
We apologize for the ambiguity. We have reformulated the sentence as follows: “Associations were confirmed between larger step length, range of motion in the hip and knee flexion/extension on the one side and longer distances in the 6MWT, faster walking speed in the 10MWT, less time needed in the TUG and moderate relations with increase in strengths of the affected lower limb, as measured with the MI-LE, on the other side.”
the results from intra/inter-person variation could be interesting.
Thanks for the remark.
L 445 – it would be useful to discuss the possible effects (eg. sex, age) here.
We appreciate this recommendation and included the following section: “Research on a larger population, allowing to systematically explore the effect of age, gender, assistive devices and gait pathologies on different severity levels is recommended to enable a balanced dataset and the formation representative subgroups.”
Very clear conclusions.
Thank you.
Reviewer 4 Report
The manuscript is well written and full of details. All the sections are well discussed and all parts are easy to read/understand by the reader. The topic is in line with the current interest on the application of wearable devices in clinical evaluation of gait impairments. Results are well presented (really good tables and graphical pictures) and completely discussed in the discussion section. Here some small suggestions:
Line 20: Spelling error. “the aim of the study was”
Line 25 and line 59: “step length and step width”
Line 90: “People post-stroke… were recruited…”
Line 100: add a picture of a subject wearing the IMU system.
Line 130: “foot” instead of “food”
There is a large heterogeneity among participants (age, weight, assistance). What about create different groups and compare results, as additional investigations?
Figure 2: graphs of the same parameter for AS and LAS must have the same axis dimension.
Author Response
Reviewer 4
The manuscript is well written and full of details. All the sections are well discussed and all parts are easy to read/understand by the reader. The topic is in line with the current interest on the application of wearable devices in clinical evaluation of gait impairments. Results are well presented (really good tables and graphical pictures) and completely discussed in the discussion section. Here some small suggestions:
Thanks for the positive feedback and provision of suggestions.
Line 20: Spelling error. “the aim of the study was”
Thanks for pointing out this error. The wording has been adapted.
Line 25 and line 59: “step length and step width”
The spelling errors were corrected.
Line 90: “People post-stroke… were recruited…”
The wording has been adapted according to the reviewer’s suggestion.
Line 100: add a picture of a subject wearing the IMU system.
Thanks for the suggestion. Unfortunately, we do not have a photo of a subject wearing the IMU system. For illustration of the sensor system, we have included a set-up schematic in Figure 1.
Line 130: “foot” instead of “food”
The error has been corrected.
There is a large heterogeneity among participants (age, weight, assistance). What about create different groups and compare results, as additional investigations?
We appreciate the reviewers recommendation. We have considered the effect of age, weight and height in our preliminary analysis. We did not find any correlation between these factors and the gait metrics. As partially discussed, further subgrouping based on assistance was not considered based on small and imbalanced subgroups.
Figure 2: graphs of the same parameter for AS and LAS must have the same axis dimension.
Thanks for highlighting this inconsistency. The axis dimensions were adapted accordingly in Figure 2(a) and (b).
Round 2
Reviewer 1 Report
All the issues are solved except the reference 67 and 68. Please correct it.
Author Response
Thanks for the remark. The references were corrected.
In addition, the English language editing of the manuscript is completed.